# Do metacognitions contribute to pathological health anxiety? A systematic review and meta-analysis

**Lavinia Ivan**[1], **Petra Foerster**[1], **Frederic Maas genannt Bermpohl**[2],
**Alexander L. Gerlach**[1], **Anna Pohl**[1]*

**1** Department of Clinical Psychology and Psychotherapy, University of Cologne, Cologne, Germany,
**2** Department of Clinical Psychology and Psychotherapy, University of Wuppertal, Wuppertal, Germany

\* anna.pohl@uni-koeln.de

## Abstract

### Objective

The purpose of this meta-analysis is to give an overview of the relationships between positive and negative metacognitions (PMC, NMC) with health anxiety and pathological safety seeking and avoidant behavior (SSB, AB).

### Method

The preregistered systematic literature screening included following data bases: MEDLINE, PsycINFO, PSYNDEX, PubMed, The Cochrane Library, Web of Science, ProQuest Dissertations & Theses, and The German National Library. The studies were evaluated based on predefined eligibility criteria (i.e., data for PMC/NMC and health anxiety and/ or SSB/AB from adult samples, assessed with validated inventories and presented in English or German language) and risk of bias categories. Correlation coefficients were aggregated with random effect models. Publication biases were estimated with contour enhanced funnel plots and outlier analyses.

### Results

23 studies ($N = 9229$) were included in the meta-analysis. Most studies assessed health anxiety in analogue samples. A significant medium effect was found for PMC and health anxiety ($r = .36$, $p < 0.0001$, 95% CI: $.29 \leq r \leq .43$), whereas for NMC the effect was large ($r = .52$, $p < 0.0001$, 95% CI: $.46 \leq r \leq .58$). For the relationship with SSB the results revealed a moderate effect for PMC ($r = 0.31$, $p = .004$; 95%-CI: $0.19 \leq r \leq 0.42$) and a small effect for NMC ($r = .25$, $p = .02$, 95% CI: $.05 \leq r \leq .43$). No study assessed AB.

### Discussion

Metacognitions are a significant pathological factor in health anxiety, with particularly strong association with NMC. PMC might be of special interest for health anxiety and

**Data availability statement:** The R dataset and the full R script are available on OSF (https://osf.io/yt9ds/).

**Funding:** The author(s) received no specific funding for this work.

**Competing interests:** The authors have declared that no competing interests exist.

SSB compared to other psychopathologies. Heterogeneity, missing clinical samples and studies on AB limit generalizability. Future research should further explore the role of metacognitions in health anxiety and focus on the relation with pathological SSB and AB.

## Introduction

Metacognitive beliefs refer to an individual's cognitive evaluations and beliefs about their own thoughts and thinking processes [e.g., 1]. Maladaptive metacognitive beliefs play a significant role in the onset and exacerbation of various mental disorders [2,3]. First being established for generalized anxiety disorder (GAD), Wells metacognitive model assumes that attentional biases, perseverative thinking (i.e., worrying and ruminating) and dysfunctional coping form a cognitive-attentional syndrome that manifests in mental disorders [4,5]. In contrast to traditional cognitive models, thought process regulation by metacognitive beliefs lies at the heart of this framework, while thought content is considered as subordinate (i.e., the specific worry, for example 'What if I lose my job?'). Contradictory positive metacognitions (PMC) and negative metacognitions (NMC) are held by people with pathological worries. PMC on the helpfulness of worrying initiate perseverative thinking. As an example, a person might think that worrying helps with problem solving, motivates, prepares for eventual catastrophes, or prevents harm. Whenever a specific worry is triggered, this person will continue to worry, because of those PMC. Engagement in perseverative thinking then activates NMC about the controllability and harmfulness of worries. When thoughts are experienced as particularly uncontrollable, people tend to engage in dysfunctional coping strategies, e.g., try to suppress their thoughts, or try to avoid triggers. As a result, they are more likely to remain stuck in perseverative thinking processes than people who attribute less importance to their thoughts. In other words, those strategies rather reinforce worrying and the perception of uncontrollability of worries (i.e., NMC). Various studies demonstrated that especially negative metacognitive beliefs relate to pathological worry in the context of GAD [5–8] and predict the duration, intensity, and burdensomeness of subsequent worry episodes [9,10], as well as generalized anxiety longitudinally [2].

In addition to their relevance for GAD, metacognitions are supposed to specifically impact mental disorders that are characterized by any kind of intrusive thoughts [11]. In line with this, besides GAD, especially large effect sizes of NMC were found for obsessive compulsive disorder and major depression when contrasting different psychopathologies meta-analytically [2]. Moreover, metacognitive therapy has been proven to be an effective transdiagnostic therapy, superior to standard cognitive behavioral therapy [for a meta-analysis see 12].

Although intrusive thoughts on health threats are common in pathological health anxiety [11,13,14], research on metacognitions in this condition remains underrepresented [e.g., being not included in the meta-analysis of Sun et al., 2]. This is despite

the early adaptation of the metacognitive model to pathological health anxiety and its shared core features with GAD and obsessive-compulsive disorder (OCD). Pathological health anxiety is diagnosed when individuals are preoccupied with having or becoming a severe disease [15] and in its severe form a significant individual and societal burden in the long-term [16]. Due to uncertainty concerning their health status, individuals may engage in excessive information sampling (e.g., checking the body, reassurance seeking, googling). Conversely, they might avoid any information that triggers their health anxiety (e.g., conversations about medical conditions, medical examinations, thinking about being ill). Safety seeking behaviors and avoidant behaviors are the two classes of dysfunctional behaviors typically maintaining pathological health anxiety. According to the *Diagnostic and Statistical Manual of Mental Disorders* (DSM 5), individuals can be categorized as either care-seeking or care-avoidant [15]. However, most patients engage in both dysfunctional behaviors, and intra- and interindividual fluctuations remain poorly understood [17]. Currently, in both research and practice, the cognitive behavioral model [18] predominates the field, and most treatment approaches are based on this model. Although some early descriptions of cognitive models already include cognitions that could be conceptualized as metacognitions [for example in 18, PMC: "If you don't go to the doctor as soon as you notice anything unusual then it will be too late.", p.110], the treatment focus tends to be on questioning specific cognitions relevant to the content of the illness. However, some authors argue that metacognitions may play a more important role compared to specific dysfunctional beliefs [19], making them an interesting target for researching pathological health anxiety.

In line with research on other psychopathologies, larger effects were reported for the association of NMC and health anxiety compared to the positive counterpart [e.g., 20]. Similarly, the assumption about the uncontrollability and interference of illness thoughts seems to play an important role [21], and may represent a predictive factor for health anxiety [11,22,23].

However, research results to date are heterogeneous, with effects for PMC ranging from null effects [11] to large associations [24]. Moreover, a recent study on mechanisms of cognitive behavioral and metacognitive therapy reported that treatment-induced changes in positive and not negative metacognitions predicted subsequent anxiety [25].

It has been suggested that the differentiation of PMC and NMC helps identify cognitions that foster specific maintaining behaviors in pathological health anxiety [20]. Investigating an analogue sample, the authors found that PMC correlated with medical consultations, whereas NMC had a strong association with depression. A possible interpretation for this result is that PMC (e.g., "I can protect myself against getting a serious disease by thinking the worst about symptoms.") might increase safety seeking behaviors (SSB).

In contrast, NMC might be more related with avoidance behavior (AB). For example, Weck et al. [26] postulate that SSB serves to reduce or cope with health anxiety whereas AB is utilized to prevent the onset of health anxiety. Metacognitive beliefs that the preoccupation with having a serious disease is uncontrollable and overwhelming may therefore, increase the fear that illness anxiety could anytime be triggered by any internal or external stimulus.

A first systematic review evaluating the role of metacognitive beliefs in somatic distress was published by Keen et al. [27]. In this review, twelve studies covered health anxiety and medium to large correlations with metacognitive beliefs were described. A meta-analytical evaluation of the effects and a clear differentiation of averaged effects for PMC and NMC concerning uncontrollability of thoughts were beyond the scope of the systematic review. Moreover, there is no review or meta-analysis describing the state of research concerning the association between positive and negative metacognitive beliefs and safety seeking and avoidant behavior in the context of health anxiety. Accordingly, the focus of this review and meta-analysis is to investigate the associations between PMC and NMC, health anxiety, safety-seeking and avoidant behavior in order to gain a better understanding of the factors contributing to pathological health anxiety in adults. We assume that PMC and NMC are positively related to health anxiety, whereby the effect for PMC might be more heterogeneous. Furthermore, PMC might be more strongly related with SSB, while NMC might specifically foster AB.

## Method

This meta-analysis was preregistered with PROSPERO (ID: CRD42023471335). It was conducted in accordance with the PRISMA 2020 guidelines (see S1 Table for the completed PRISMA checklist). The R dataset and the full R script are available on OSF (https://osf.io/yt9ds/?view_only=0c4607b006ca4523b471a6f7e2c24653).

### Literature search

The following 8 databases were searched by one researcher (P.F.) on the 27.11.2023: MEDLINE, PsycINFO, PSYN-DEX, PubMed, The Cochrane Library, Web of Science, ProQuest Dissertations & Theses, and The German National Library. The following terms were searched: ((metacogn* OR "meta-cogn*") AND (hypochondr* OR "illness anx*" OR "health anx*" OR cyberchondr* OR "disease phobia" OR "health worr*")) OR ((metacogn* OR "meta-cogn*") AND (safety* OR reassurance* OR healthcare * OR "medical utilization" OR "illness behavior" OR "illness behaviour" OR scanning OR "body checking" OR "body-checking" OR care-* OR checking* OR avoid* OR "information avoid*")). The exact documentation of the searches (e.g., specific characteristics in certain databases) can be found in S2 Table.

### Study selection

After applying the search strategies, the first author (L.I.) uploaded the results to the internet-based software program RAYYAN QCRI [28]. In total, $n = 6050$ reports were identified through the searches. Two researchers (L.I. and P.F.) deleted duplicates ($n = 2632$), reviewed the titles and abstracts independently, and inspected eligible full texts. If the full text was not available ($n = 3$), the corresponding author was contacted once via email. Only (a) published or unpublished studies (e.g., dissertations) in English and German, implementing (b) quantitative designs that (c) reported data at least for metacognitive beliefs and health anxiety or safety-seeking/avoidant behavior in the context of health anxiety were considered. Studies were excluded when constructs were not assessed with validated inventories. To increase the power of our meta-analysis, we deviated from the preregistration and included two studies, which assessed 16–18 years old participants next to adults, but had a similar age average as other analogue samples. Reference lists of the included studies and meta-analyses ($n = 9$) in the field were checked. The reasons for exclusion were documented in RAYYAN QCRI [28]. A compilation of excluded studies which might appear to meet inclusion criteria is available in the supplement (S2 Table). If questionnaires were modified from the original version ($n = 1$), the study was initially included and then influence analyses were conducted to check for result bias. Multiple publications of the same data [$n = 1$, 21,29] were treated as one study [30]. Disagreement over selection at any point was discussed with a third researcher (A.P., $n = 15$) to reach consensus.

### Data extraction

Two researchers (L.I. and P.F.) independently extracted the following relevant data from each study using a pre-designed Microsoft Excel Sheet: authors, publication date, sample size, participant characteristics (sex, age, analogue sample, or clinical sample with control group, education), measures used to diagnose health anxiety, recruitment method, inventories used to assess constructs of interest, statistical analysis, subgroups, effect sizes for the relations of interest (correlation coefficients) or data to calculate effect sizes. If multiple measures for metacognitive beliefs were reported, health anxiety-specific instruments were preferred. For the other constructs, the data from the most common instrument were extracted to reduce possible heterogeneity. The researchers compared their results, discussed any disagreements, and involved a third researcher (A.P.), if necessary. If relevant statistical information was missing, the corresponding author was contacted via e-mail and asked to provide the article or missing data (one reminder after one week).

## Risk of bias

To assess risk of bias, criteria from a JBI-checklist [31] and items from the risk of bias (ROB) statement by Von Elm et al. [32] were combined. We evaluated the studies considering seven domains: preregistration, power analysis, description of sample, selection of participants, reliability, replicability, and statistical methods. In total, 19 assessment criteria were rated using three categories: ROB, no ROB and not applicable (N/A). The assessment was conducted by two study investigators independently (L.I. and P.F.) and any conflict ($k = 6$) was discussed or resolved with a third researcher (A.P.). The ROB for each included study (for all criteria individually and in total), as well as an aggregated result for each criterion across all studies is reported. The interrater reliability (Cohen's κ) was calculated across all domains.

## Data preparation

Regression coefficients instead of correlation coefficients were reported in one study [33]. The three extracted standardized regression coefficients were transformed into correlation coefficients with the 'esc_beta' function in R. For studies which used an instrument with more than one relevant subscale, or which reported the correlations separately for subgroups [$k = 3$, e.g., participants in good vs. poor health, 20], the corresponding correlation coefficients were averaged using Fisher's $z$-transformation and weighted based on sample sizes (see uploaded data and scripts). Details of the questionnaires used, and the pooled subscales are available in S4 Table.

## Data synthesis

Statistical analyses were performed using R and R Studio and the {metafor} package [34]. Random effects models with a Knapp-Hartung adjustment were used to aggregate the effect sizes. Separate analyses were calculated for PMC and NMC, as well as health anxiety, SSB and AB. Effects sizes were visualized with forest plots. To specify and improve the robustness of the models, we regarded the following aspects:

**Heterogeneity.** Because of various concerns and weaknesses of the different methods used to examine heterogeneity, multiple statistical approaches were employed. The $Q$ statistic and the $\tau^2$ value were examined. The $I^2$ statistic [35] was calculated to examine and quantify heterogeneity [low: $< 25$, moderate: 25–75, or high: $\geq 75$, 36]. To identify the studies, which contribute to the heterogeneity, an outlier and influence analysis was performed. To identify influential studies, the Baujat plot, influence diagnostics according to Viechtbauer & Cheung [37] and the leave-one-out analysis, using the 'InfluenceAnalysis' function of the {dmetar} package were used. If there were influential studies in the meta-analysis, we report results of the sensitivity analysis in which these influential cases were excluded. To detect outliers and (if necessary) remove them and recalculate the results (including $I^2$) the 'find.outliers' function of the {dmetar} package was used.

**Publication bias.** There is no ideal method for assessing publication bias. For this reason, several methods [35] were combined. Contour-enhanced funnel plots combined with the Egger's regression test were used to examine small-study effects. Complementary the $p$-curve method which focuses on $p$-values was utilized to test for evidential value.

**Moderator analyses.** In case of heterogeneity, we planned subgroup analyses for the following factors: (a) clinical sample vs. analogue sample, (b) measures of metacognitive beliefs (MCQ-HA [Metacognitions Questionnaire – Health Anxiety] vs. MCQ-30/MCQ-65), (c) measures of health anxiety (WI/WI-6 vs. SHAI/SHAI-14), and (d) statistical outcome (singular correlation vs. averaged correlation). Subgroup analyses were only performed if $k \geq 3$ in each subgroup [38]. Additionally, we included the number of categories with no ROB as a continuous factor in the moderation analysis. We calculated the effect sizes for each group and examined whether the effect sizes differed significantly from each other. A random effects model was used to pool the effect sizes of the subgroups and a fixed effects model to test for significant differences. To test if the groups differ significantly, the $Q$-test was calculated. Also, the $I^2$ statistic was calculated as well as the mean effect size (plus confidence interval and $p$-value) for each subgroup.

## Results

### Study selection

After the title-abstract screening, there were 51 studies sought for retrieval. Having access to 48 abstracts only, the authors of 3 studies were contacted to provide their full texts. One study was still under review, another study was never published and the authors of the third study did not answer. There were 14 full texts with missing or incomplete data reports. The authors of 5 studies sent us the relevant information. Concerning the other studies, the data were not accessible anymore ($n=4$) or the researchers did not reply ($n=5$). Finally, a total of 23 published studies were included in the meta-analysis. The PRISMA flow diagram [39] is shown in Fig 1. For an overview of the included studies see Table 1. Studies that were included in the meta-analysis but are not discussed in the paper can be found in the supplement (S1 Text).

### Study characteristics

All studies measured metacognitive beliefs ($k=1$, 4.34% only NMC) and health anxiety, whereas only $k=7$ (30.43%) studies included SSB, and no study assessed AB. For assessing metacognitive beliefs, the MCQ-HA [24] was used most frequently ($k=12$, 52.17%) and for health anxiety the WI [55] or the shorter version WI-6 [56, $k=13$, 56,2%]. To measure SSB, the research teams asked the participants for the amount of different physician visits during the last year ($k=1$, 4.34%),

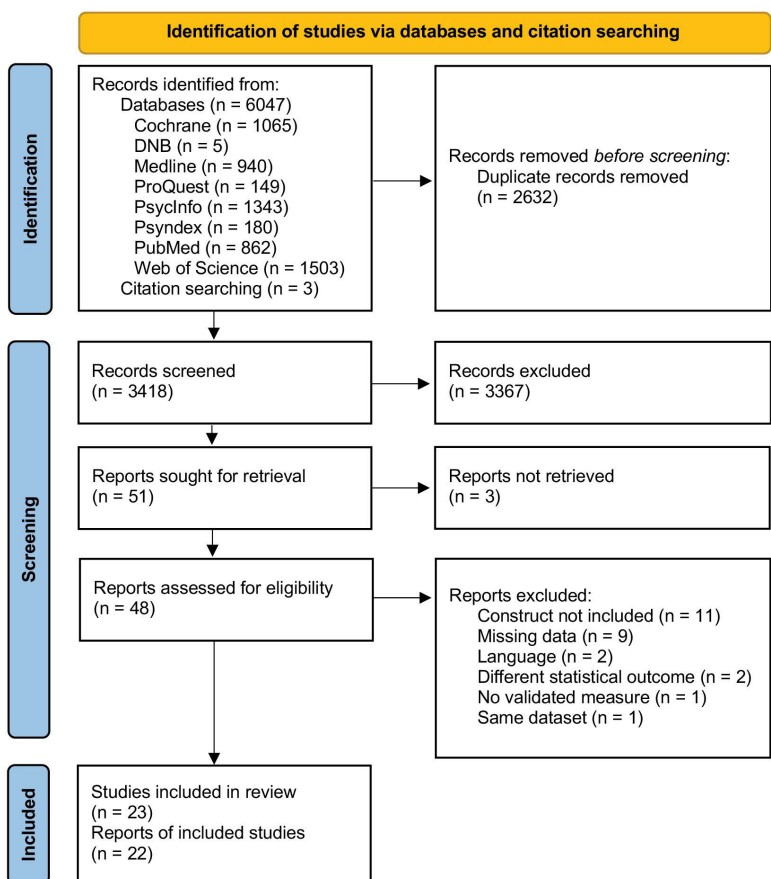

**Fig 1. PRISMA 2020 flow diagram of study selection, adapted from Page et al. [39].**

**Table 1. Description of included studies.**

| Nr. | Author and year | | N | Age M (SD) | Gender (female) | Sample | PMC | NMC | HA | SSB |
|---|---|---|---|---|---|---|---|---|---|---|
| 1 | Airoldi et al. 2022 [40] | | 125 | 34.51 (14.08) | 58.40% | analogue | MCQ-HA | MCQ-HA | SHAI-14 HCQ | CSS |
| 2 | Akbari et al. 2021 [41] | | 541 | 41.3 (13.2) | 52.30% | analogue | MCQ-30 | MCQ-30 | SHAI | NA |
| 3 | Bailey & Wells 2013 [29] | | 351 | 27 (7.48) | 89.50% | analogue | MCQ-30 | MCQ-30 | WI | NA |
| 4 | Bailey & Wells 2015 [24] | | 259 | 26 (6.9) | 91.00% | analogue | MCQ-HA MCQ-30 | MCQ-HA MCQ-30 | WI | NA |
| 5 | Bailey & Wells 2016 [33] | | 105 | 26 (6.52) | 72.40% | analogue | MCQ-HA | MCQ-HA | WI HCQ | NA |
| 6 | Barenbrügge et al. 2013 [20] | | 1264 | 44.5 (13.9) | 64.50% | analogue | WW-H[1] | MKF-30[2] | WI | number of different doctor visits |
| 7 | Bouman & Meijer 1999 [11] | | 161 | 35.4 (9) | 82.61% | both | MCHA MCQ-65 | MCHA MCQ-65 | WI | NA |
| 8 | Cartwright-Hatton & Wells 1997 [42] | | 104 | 26 (NA) | 54.81% | analogue | MCQ-65 | MCQ-65 | AnTI | NA |
| 9 | Dai et al. 2018 [43] | | 1191 | 19.33 (1.32) | 61.00% | analogue | MCQ-HA MCQ-30 | MCQ-HA MCQ-30 | SHAI | NA |
| 10 | Fergus & Bardeen 2019 [44] | | 785 | 35.3 (11.1) | 52.60% | analogue | MCQ-30 | MCQ-30 | WI-6 | NA |
| 11 | Fergus & Spada 2017 [45] | | 260 | 32.9 (9.2) | 40.80% | analogue | MCQ-HA | MCQ-HA | WI-6 | CSS |
| 12 | Fergus & Spada 2018 [46] | study 1 | 330 | 19.4 (2.1) | 66.60% | analogue | MCQ-HA | MCQ-HA | WI-6 | CSS |
| 13 | | study 2 | 331 | 38.7 (10.4) | 53.50% | analogue | MCQ-HA | MCQ-HA | WI-6 | CSS-15 |
| 14 | Fergus et al. 2022 [47] | | 307 | 35.87 (14.56) | 88.93% | analogue | MCQ-HA | MCQ-HA | WI-6 | NA |
| 15 | Kaur et al. 2011 [22] | | 158 | 18.7 (1.16) | 72.80% | analogue | MCHA MCQ-30 | MCHA MCQ-30 | WI | NA |
| 16 | Melli et al. 2018 [48] | | 458 | 33.97 (12.18) | 67.00% | clinical | MCQ-HA | MCQ-HA | HAQ HCQ | NA |
| 17 | Melli et al. 2016 [23] | | 342 | 37.69 (12.2) | 61.40% | analogue | MCHA | MCHA | HAQ HCQ | NA |
| 18 | Nadeem et al. 2022 [49] | | 500 | NA | 51.20% | analogue | MCQ-HA | MCQ-HA | SHAI-14 | CSS |
| 19 | Penney et al. 2020 [50] | | 565 | 21.46 (5.66) | 76.80% | analogue | MCQ-30 | MCQ-30 | SHAI | NA |
| 20 | Rachor & Penney 2020 [51] | | 179 | 22.18 (5.41) | 81.56% | analogue | MCQ-HA | MCQ-HA | SHAI | NA |
| 21 | Solem et al. 2015 [52] | | 382 | 26.15 (5.72) | 44.8% | analogue | MCQ-30 | MCQ-30 | WI | NA |
| 22 | Wells & Papageorgiou 1998 [53] | | 105 | 21.1 (4.14) | 63.81% | analogue | MCQ-65 | MCQ-65 | AnTI | NA |
| 23 | Zheng et al. 2021 [54] | | 426 | NA | 60.30% | analogue | | MCQ-HA[2] | WI-6 | OHIS CSS-12 |

Abbreviations: PMC = positive metacognitive beliefs, NMC = negative metacognitive beliefs, HA = health anxiety, SSB = safety-seeking behavior, AB = avoidance behavior, MCQ-HA = Metacognitions Questionnaire-Health Anxiety, MCQ-30 = Metacognitions Questionnaire-30 (30 Items), MCQ-65 = Metacognitions Questionnaire-65 (65 Items), WW-H = Why do people worry about health?, MCHA = Metacognitions about Health Anxiety, MKF-30 = Metacognitions Questionnaire-30 (short form in German, items adapted illness-specifically), SHAI-14 = Short Health Anxiety Inventory-14 (14 Items), SHAI = Short Healthy Anxiety Inventory (18 Items), WI = Whiteley Index (14 Items), WI-6 = Whiteley Index (6 Items), HCQ = Health Cognitions Questionnaire, AnTI = Anxious Thought Inventory, HAQ = Health Anxiety Questionnaire, CSS = Cyberchondria Severity Scale (33 Items), CSS-12 = Cyberchondria Severity Scale-Short Form (12 Items), CSS-15 = Cyberchondria Severity Scale-15 (15 Items), OHIS = online health information seeking assessed with 3 items. [1] Modifications of the original scale. [2] Only the subscale "beliefs that thoughts are uncontrollable" was assessed.

adjusted three items from Griffin et al. [57] to measure online health information seeking (OHIS, $k = 1$, 4.34%), or used different versions of the Cyberchondria Severity Scale [58, $k = 6$, 26.08%].

## Sample characteristics

In total, 9229 participants were included in 23 studies. The average age of the participants was 31.25 years ($n = 8303$, $SD = 8.72$, range: 16–80 years, $n = 7372$). Two studies did not include the average age, 7 studies did not report the age range. 5903 women (64%) and 3316 men (36%) participated. Twenty-one (91.30%) studies examined an analogue

sample, one study (4.35%) recruited a clinical sample [self-reported hypochondriasis or illness anxiety disorder, 48], and one study (4.35%) assessed both [a clinically diagnosed sample, 11]. The average level of health anxiety varied between the studies ranging from low to severely health anxious on the respective self-report scales. Often averaged health anxiety was mild. All mean values and standard deviations for the included studies can be found in the supplementary S5 Table.

### Risk of bias

The detailed ROB analysis for each study can be found in the Supplement (S6 Table). Across studies, there was relatively low ROB concerning the replicability and the description of the samples, except that reports of educational background were lacking in more than half of the studies. A high ROB across studies was found concerning power analyses (e.g., $k = 19$, 82.61% of the studies did not report any kind of power analyses), selection of participants (e.g., $k = 15$, 65.22% of the studies did not specify dropout or exclusion rates) and statistical methods (e.g., $k = 14$, 60.87% of the studies did not report if missing data were existent). None of the studies was preregistered. The interrater reliability across all domains was excellent ($κ = 0.93$).

### Effect sizes for positive and negative metacognitive beliefs and health anxiety

All studies examined the relationship between metacognitive beliefs and health anxiety. While $k = 1$ study reported correlations for negative metacognitive beliefs and health anxiety, $k = 22$ studies considered both NMC and PMC. The random effects analysis for PMC showed a significant correlation with health anxiety with an estimated $r = .36$ ($p < .0001$; 95%-CI:.29;.43). The estimate of the variance in true effects was $τ^2 = 0.03$ (95%-CI:.02;.06) and indicated the existence of between-study heterogeneity. The $Q$-Test of heterogeneity was significant with $Q(21) = 272.76$, $p < .0001$. Total heterogeneity accumulated to $I^2 = 92.3\%$ (95%-CI: 89.6; 94.3). For NMC, the random effects analysis revealed a significant positive correlation with health anxiety with an estimated $r = .52$ ($p < .0001$ 95%-CI:.46;.58). There were also indicators of high between-study heterogeneity with $τ^2 = 0.03$ (95%-CI: 0.02; 0.07), a significant $Q$-Test with $Q(22) = 364.61$, $p < .0001$ and $I^2 = 94.0\%$ (95%-CI: 92.1; 95.4). For the exact data see the forest plots in Fig 2A and 2B.

   **Sensitivity analyses.** Random effect models without outliers revealed a marginal increased effect for the relation with PMC with reduced but still high heterogeneity. Concerning NMC, the effect without outliers remained the same and heterogeneity decreased to a moderate level (detailed statistics, Baujat Plots and leave-one-out analyses can be found in S1 Appendix).

### Effect sizes for positive and negative metacognitive beliefs and safety seeking behavior in the context of health anxiety

The behavioral component of health anxiety was measured in $k = 7$ studies. Only $k = 4$ studies reported the relevant correlations for the relation of SSB and PMC and $k = 5$ studies the correlations for SSB and NMC. The random effects analysis for PMC showed a significant positive correlation between PMC and SSB, $r = .31$ ($p = .004$; 95%-CI:.19;.42). The heterogeneity indicator $τ^2 = .004$ (95%-CI: 0.00; 0.08) suggests that there is no heterogeneity, whereas the $Q$-Test: $Q(3) = 8.43$, $p = .04$ was significant and $I^2 = 64.4\%$ (95%-CI: 0.0; 88.0). The random effects analysis for NMC also revealed a significant positive correlation between NMC and SSB with an estimated $r = .25$ ($p = .02$; 95%-CI:.05;.43). There were indicators of high between-study heterogeneity with $τ^2 = 0.02$ (95%-CI: 0.01; 0.21), a significant $Q$-Test with $Q(4) = 50.96$, $p < 0.0001$ and $I^2 = 92.2\%$ (95%-CI: 84.7; 96.0). For the exact data see the forest plots in Fig 3A and 3B. For SSB no outliers were detected in sensitivity analyses.

### Publication bias

Eggers' test revealed no significant indication of funnel plot asymmetry for the correlations between health anxiety and either PMC ($p = .369$) or NMC ($p = .095$; see S1 Fig for the contour-enhanced funnel plots). Due to the limited number of studies, no formal test was conducted for relation with SSB. Nonetheless, the corresponding funnel plots are included in the

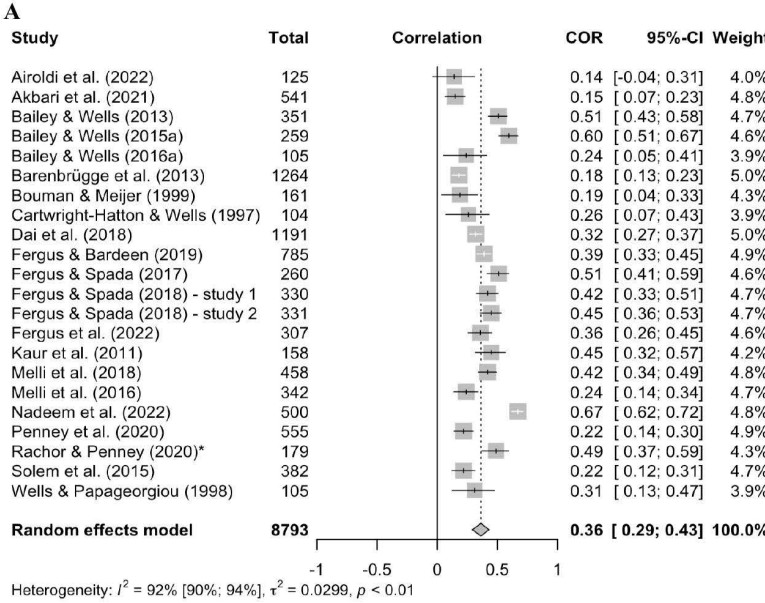

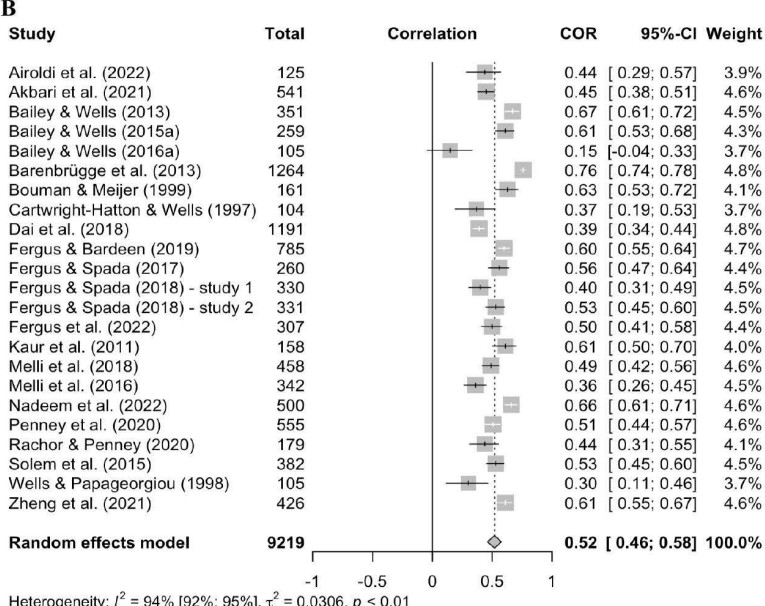

**Fig 2. Forest Plots for the relation of health anxiety and positive (A) as well as negative metacognitive beliefs (B).** Abbreviations: COR = correlation coefficient, CI = confidence interval, *multiplied with '-1' due to an inverse transformation of the PMC scale.

Supplement for reference (S1 Fig). Evidential value was present for the relation of both PMC and NMC with health anxiety and SSB respectively in the *p*-curve analyses. The corresponding *p*-curves are provided in the Supplement (S2 Fig).

## Moderator analyses

Effects for clinical vs. analogue samples (a) could not be compared, because only $k = 2$ studies reported separate correlations for the clinical subgroup [11,48]. The relationship between metacognitive beliefs (MCs) and health anxiety did not

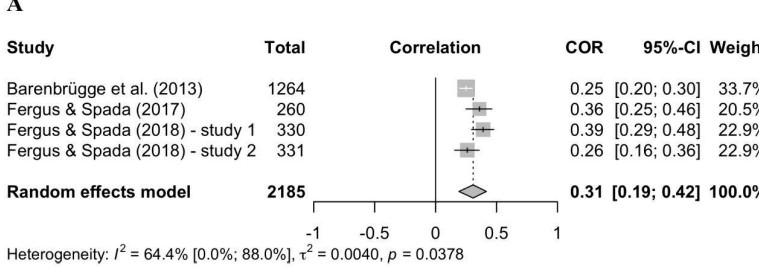

**A**

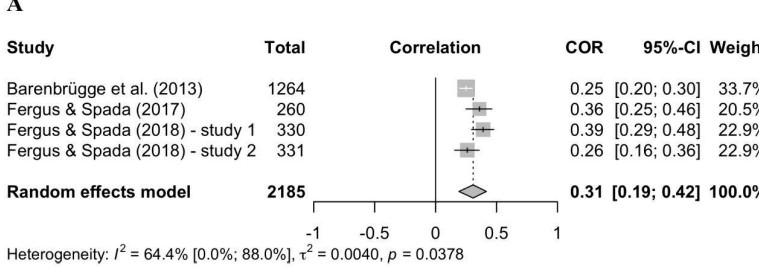

**B**

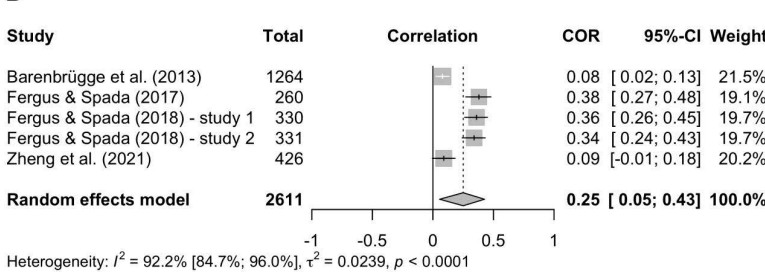

**Fig 3. Forest Plots for the relation of safety-seeking behavior and positive (A) as well as negative metacognitive beliefs (B) in the context of health anxiety.** Abbreviations: COR = correlation coefficient, CI = confidence interval.

differ significantly when comparing health anxiety-specific metacognitive questionnaires with general ones, whereby there was a trend for a higher relation of health anxiety specific questionnaires in PMC ([b] MCQ-HA vs. MCQ-30/MCQ-65). No significant difference was found for different assessments of health anxiety ([c]WI/WI-6 vs. SHAI/SHAI-14). Subgroup analyses did not reveal significant differences between studies with singular correlations and those with averaged correlations (d). For detailed statistics, refer to S7 Table.

The number of categories with no risk of bias (ROB) per study did not significantly predict the relation of health anxiety with PMC ($p = .71$) or NMC ($p = .50$). Similarly, the associations of SSB and health anxiety were not related to the number of no ROB categories (PMC: $p = 0.97$; NMC: $p = 0.76$).

## Discussion

### Relation between metacognitions and health anxiety

For the relationship between PMC and health anxiety, a significant medium effect was found, whereas for NMC and health anxiety the effect was large [59]. Both effects support the results of a recent narrative review statistically, based on a database nearly twice as large [27]. The association of NMC and health anxiety is well in line with meta-analytical reports of strong correlations in GAD and OCD, whereby effects for health anxiety were descriptively slightly smaller in the current meta-analysis [compare 2]. These results underline the impact of NMC concerning the uncontrollability of thoughts. Consistent with Wells' metacognitive model, NMC play a central role in several different mental disorders [2], as well as in anxiety and depression in the context of multiple physical illnesses [60]. Note, however, that other metacognitions might be more specific for certain physical illnesses [e. g., positive beliefs about worry, 60].

We found numerically larger effects for the relation of PMC and health anxiety compared to previous results in GAD and OCD [2], as well as anxiety and depression in physical illnesses [60]. Furthermore, our analyses revealed a moderate positive relation for SSB and PMC, while only a small positive effect for NMC was found, with studies being more heterogenous for NMC. Interestingly, while NMC had previously been shown to be related to all three subfacets (emotional

'Disease Phobia' and cognitive 'Bodily Pre-Occupation', 'Conviction of the Presence of Disease') of the Whiteley Index, PMC were especially associated with the 'Disease Phobia' subscale of this inventory [20]. PMC were thus associated with the emotional component of illness anxiety as well as SSB, a dysfunctional strategy to reduce anxiety, or uncertainty concerning the possibility of getting or being seriously ill. As hypothesized, this points to the particular relevance of PMC for health anxiety.

Of the five studies assessing the relation of MCs and SSB, four operationalized self-reports of (excessive) online information seeking as SSB. In one study participants were asked to rate the number of different physicians visited in the last year. While the relation of SSB with NMC and PMC was similar when assessed with the subscales 'Excessiveness' and 'Reassurance' of the Cyberchondria Severity Scale [45,46], asking for the number of different physicians visited in the last year [20] yielded a somewhat lower relation with NMC. That means, although reassurance was assessed in both instances, the query of actual behavior within a specified time period might make a difference. Furthermore, a small relation of NMC and SSB was reported when online health related information was not assessed with the Cyberchondria Severity Scale [54].

Our meta-analysis revealed a lack of research concerning the associations between the behavioral component of health anxiety and MCs. Research indicates intra- and interindividual fluctuations of safety seeking and avoidance behavior [17,61,62], with the majority of individuals characterized by pathological health anxiety performing both behaviors [17]. Nevertheless, SSB was rarely researched, and we found no empirical study on the relation of AB and MCs. This is in line with a recent meta-analysis on the experimental assessments of pathological mechanisms in health anxiety, in which 3 out of 57 studies addressed SSB and no study examined AB [63]. This gap in research is striking, given that AB is considered a highly relevant maintenance factor in models of pathological health anxiety [64,65] and has a strong theoretical connection with NMC [66].

### Heterogeneity, influencing factors and risk of bias

The heterogeneity of the included studies was substantial and could not be explained by preregistered moderators. Some moderators could not be assessed due to an insufficient amount of studies (e.g., clinical vs. analogue sample, singular vs. averaged correlation in case of PMC), others did not explain the heterogeneity. Contrary to previous assumptions suggesting a more nuanced comprehension of the metacognitive model through the utilization of health anxiety-specific questionnaires [27], it was found that these specific metacognitive questionnaires were not more closely associated with health anxiety than the general ones concerning NMC. However, this finding contrasts with previous studies, which found incremental variance explanation of health anxiety specific over the non-specific metacognitive questionnaires [11,24,43]. Arguably, the additional variance may be less relevant than previously assumed. However, merging of the general inventories MCQ-65 and MCQ-30 may also have masked differences between the specific and general inventories, although factorial structure and validity of the short form seem a good representative of the original MCQ-65 [67–69]. Furthermore, there was a trend for PMC, which should be replicated with a larger data base. Similarly, the measures of health anxiety did not influence its relation with MCs. This is in line with studies finding that different health anxiety inventories [e.g., WI and (S)HAI, 70–73] often exhibit similarly high correlations, as do adaptations of any inventory [e.g., WI vs. WI-6, 56]. Finally, we found no significant influence of the total number of categories without ROB. However, as the calculation implies an equal weighting of all categories, this result is perhaps not as meaningful as a qualitative description of the categories with increasingly high or low ROB.

Despite the large heterogeneity, effects for the relation of MCs and health anxiety can be considered robust. Outliers are distributed fairly evenly across stronger and weaker effects, with their magnitudes confirmed by sensitivity and outlier analyses. Nevertheless, consideration should be given to identifying other factors that may contribute to heterogeneity. In a large study with broad age range increased MCs were reported for participants with actual physical illness compared to participants without concurrent diagnosis. Furthermore, NMC were associated with younger age, while education and

gender had no influence on MCs [20]. Keen and colleagues [27] furthermore argued that sample composition might have impacted the relation of MCs and health anxiety since a considerable amount of studies with medium or large effects recruited medical or nursing students.

### Strength and limitations

This meta-analysis included a sufficient number of studies. The central constructs were consistently defined and assessed using comparable and validated instruments. Most of the studies had reasonably large sample sizes. However, even after conducting outlier analyses, heterogeneity remained moderate to high. The state of research regarding the relationship between MCs and SSB/AB appears to be poor and it must be taken into account that our results for associations with SSB were based on a small number of studies. Moreover, most studies recruited young analogue samples, while studies in older populations are lacking.

### Future directions

Although cognitive behavioral therapy is effective for treating pathological health anxiety, response and remission rates indicate that 50% of affected individuals do not profit to a satisfactory extent [74]. This meta-analysis provides evidence for Wells model, according to which people with increased anxiety hold both, PMC and contradictory NMC [4]. Thus, it seems plausible to include metacognitions in case conceptualizations. However, the interplay between these metacognitions and dysfunctional coping strategies has not yet been sufficiently investigated in illness anxiety. Therefore, it seems sensible to attempt to better understand the mechanisms involved in the emergence and perpetuation of health anxiety. To increase our understanding, future studies should investigate clinical samples, manipulate metacognitions [e.g., 6] and examine their influence on SSB and AB. The potentially specific role of PMC for health anxiety and SSB has to be proven in studies comparing different clinical samples and studies which link specific metacognitions to different dysfunctional strategies (SSB, avoidance, hypervigilance). Finally, it would be of interest to examine, whether metacognitions explain incremental variance beyond and above other mechanisms relevant for pathological health anxiety [75], explain SSB and AB in day to day life and predict health anxiety longitudinally.

## Supporting information

**S1 Table. PRISMA 2020 checklist.**
(DOCX)

**S2 Table. Literature search.**
(DOCX)

**S3 Table. Excluded studies which might appear to meet the inclusion criteria.**
(DOCX)

**S4 Table. Data preparation.**
(DOCX)

**S5 Table. Sample characteristics.**
(DOCX)

**S6 Table. Risk of bias and publication bias: Detailed ROB analysis.**
(DOCX)

**S7 Table. Moderator analyses.**
(DOCX)

**S1 Text. References of articles included in the meta-analysis.**
(DOCX)

**S1 Appendix. Sensitivity analyses.**
(DOCX)

**S1 Fig. Risk of bias and publication bias: Contour-enhanced funnel plots.**
(DOCX)

**S2 Fig. Risk of bias and publication bias: p-curves.**
(DOCX)

## Acknowledgments

We thank all authors, who provided us with additional data required for the analyses. We are also grateful to Hannah Korst and Moritz Schöffel, who helped with the presentation and checking of the calculations and with formatting the manuscript.

## Author contributions

**Conceptualization:** Lavinia Ivan, Anna Pohl.

**Data curation:** Lavinia Ivan, Petra Foerster, Frederic Maas genannt Bermpohl, Anna Pohl.

**Formal analysis:** Lavinia Ivan, Frederic Maas genannt Bermpohl.

**Investigation:** Lavinia Ivan, Petra Foerster.

**Methodology:** Lavinia Ivan, Anna Pohl.

**Project administration:** Lavinia Ivan.

**Software:** Frederic Maas genannt Bermpohl.

**Supervision:** Alexander L. Gerlach, Anna Pohl.

**Validation:** Lavinia Ivan, Frederic Maas genannt Bermpohl, Anna Pohl.

**Visualization:** Lavinia Ivan, Frederic Maas genannt Bermpohl.

**Writing – original draft:** Lavinia Ivan, Anna Pohl.

**Writing – review & editing:** Lavinia Ivan, Petra Foerster, Frederic Maas genannt Bermpohl, Alexander L. Gerlach, Anna Pohl.

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
