## [Decision Letter · Decision Letter 0]

PONE-D-24-39301Do metacognitions contribute to pathological health anxiety? A systematic review and meta-analysis.PLOS ONE?

Dear Dr. Pohl,

We look forward to receiving your revised manuscript.

Kind regards,

Hans-Peter Kubis, PD. Dr. rer. nat.

Academic Editor

PLOS ONE

Journal Requirements:

2. As required by our policy on Data Availability, please ensure your manuscript or supplementary information includes the following:

Reviewers' comments:

Reviewer's Responses to Questions

**Comments to the Author**

1. Is the manuscript technically sound, and do the data support the conclusions?

Reviewer #1: Partly

Reviewer #2: Yes

2. Has the statistical analysis been performed appropriately and rigorously?

Reviewer #1: Yes

Reviewer #2: Yes

3. Have the authors made all data underlying the findings in their manuscript fully available?

Reviewer #1: Yes

Reviewer #2: Yes

4. Is the manuscript presented in an intelligible fashion and written in standard English?

Reviewer #1: Yes

Reviewer #2: Yes

Reviewer #1: Review Do metacognition contribute to pathological health anxiety? A systematic review and meta-analysis

This article addresses the role of positive and negative metacognitions in health anxiety and their relationship with pathological safety-seeking and avoidant behaviors. Extending previous findings from other anxiety-related disorders, such as generalized anxiety disorder (GAD), this systematic review seeks to bridge a critical gap in understanding metacognitive contributions to health-related anxiety. The meta-analysis of the results of 23 studies showed positive correlations between Negative (NMC) and Positive Metacognition (PMC) on one hand and health anxiety, as well as with safe seeking behaviors. No included study assessed avoidant behavior.

While the methodology appears rigorous, with clear and well-detailed inclusion criteria for the selected studies, several parts of the paper lack clarity, particularly in the introduction. Additionally, inconsistencies in formatting and figures detract from the readability of the article.

Introduction

The introduction is thematically relevant but requires substantial clarification.

• The primary research question is unclear, and some terms central to the study, such as "health anxiety", "safety-seeking behaviors", and "avoidant behaviors" would benefit from explicit definitions to ensure accessibility to a broader readership such as the one of PLoS ONE.

• Although the authors cite the metacognitive and cognitive-behavioral models, these frameworks are not adequately described, making it challenging for readers unfamiliar with these concepts to contextualize the study's objectives.

• The relationship between health anxiety, safety-seeking behaviors, and avoidance behaviors is not sufficiently detailed, leaving the reader without a clear understanding of how these constructs are conceptualized or interrelated.

Methods

The methodology is robust, and the explanation of study selection and bias assessment is clear. However, some areas might require additional detail:

• How many studies reported regression coefficients and how were they transformed into correlation coefficients?

• The section on data preparation mentions subgroups, what do they refer to?

• Figure 2 should include titles for panels A and B to improve interpretability.

• For analogue samples, it might be helpful to report the average levels and variability in health anxiety

Results

The results section provides useful information but requires clarification in several places:

• In the study selection paragraph, it is unclear whether the cited sample sizes (n) are among the 23 included studies. While Figure 1 and Table 1 are clear and helpful, ensuring that the text aligns with these visuals would improve consistency.

• The introduction suggests a causal relationship and sequentiality between PMC and NMC, was the correlation between the two evaluated?

• A subgroup analysis based on gender (female/male) could provide additional insights, given the potential for gender differences in health anxiety and related behaviors.

Discussion

The discussion draws meaningful parallels with findings from related pathologies, such as GAD and obsessive-compulsive disorder (OCD), but lacks specificity regarding health anxiety.

• What are the specificities of health anxiety compared to the other cited pathologies? Exploring such differences could highlight the unique contributions of metacognitive processes in this context.

• Were the subfacets of health anxiety questionnaires analyzed in the present study?

• The authors cite Wells' metacognitive model in the introduction, how do the findings relate to this model? A more explicit connection would improve the theoretical grounding of the results.

• Even though no study assessed avoidance behavior, maybe the authors could explicit the theoretical and empirical links with health anxiety and metacognition?

Minor comments and typos

• Formats of citations are inconsistent, some are author-date and others are numbers.

• Figures are separated from their captions

In conclusion, while this article tackles an interesting and important topic, the authors should clarify their research questions, better define their objects of study, and provide a more cohesive discussion of how these constructs relate. Greater detail about the relation of the present findings with the previous ones and existing theories of health anxiety would strengthen the contribution of this review to the field.

minor issues, typos..

The formatting of references is inconsistent. Please correct it.

In several places, punctuation is incorrect, complexifying the reading

Reviewer #2: Summary:

In this systematic review and meta-analysis, the authors set out to assess associations between positive and negative metacognitions on the one hand, and health anxiety and related behaviors (safety seeking and avoidance) on the other. The study was preregistered. N = 23 studies were included in the final meta-analysis, with only few addressing safety-seeking and none assessing avoidance behavior. Results point towards medium-strong associations between metacognitions (both positive and negative) and health anxiety.

With this work, the authors provide a good and solid overview over the topic and are addressing a (surprising) lack in the literature. Their argumentation is consistently sound, and they reveal existing research gaps and point towards important future directions, both in the clinical and the research sphere.

A big strength of this work is its sound and transparent methodology, including the consistency with which the preregistration was followed. The paper is also very well written and was a true pleasure to read.

Therefore, I find the article basically ready for publication, although I would suggest fixing some super minor formulation/typo issues.

- p. 3, l. 63: replace use of “Exemplarily”

- p. 3, l. 74-76: replace (repeated) use of “next to”

- p. 16, l. 369-ff: the following sentence is perhaps a bit clunky and could be clarified: “These results underline the impact of NMC concerning the uncontrollability of thoughts, especially relevant for several different mental disorders (2) as well as for anxiety and depression across a variety of physical illnesses (57).”

 what is "especially relevant", and what is meant by "across physical illnesses" (i.e., is this part meant to refer to instances where depression and anxiety co-occur with physical illness?)

- p. 17, l. 410: missing comma before “others”?

- p. 18, l. 424: rewrite “as adaptations of anyone inventory”?

- p. 18, l. 425-ff: something seems off in the following sentence (remove "as" or add something?): “However, as the calculation implies an equal weighting of all categories, which is perhaps not as meaningful as a qualitative description of the categories with increasingly high or low ROB.”

- Supplementary/appendix: remove comment in S3 file and consider mentioning all supplements in the main text (as of now, no reference to S8 or S9, so their relevance is unclear)

- Study selection:

I was slightly confused by the information provided here as the number 23 refers to published studies, but then unpublished studies are mentioned, and it becomes also clear that not all data could be retrieved from the n = 14 studies with missing data. So, are these selections that happen before the final number of 23 for analyses is established or do these numbers concern articles included in the 23 works selected for meta-analysis? When age averages are later reported it also becomes clear that not all studies are included here, so it might be best to provide an N of both studies and participants for calculated averages.

Overall, congratulations to the authors for a great piece of work!

**Do you want your identity to be public for this peer review?** For information about this choice, including consent withdrawal, please see our Privacy Policy

Reviewer #1: No

Reviewer #2: No

---

## [Author Response · Author response to Decision Letter 1]

3 Apr 2025

Dear Dr. Kubis, dear Reviewers,

Thank you very much for your letter dated January 30th, 2025, and for the two reviews of our manuscript titled “Do metacognitions contribute to pathological health anxiety? A systematic review and meta-analysis.” (manuscript PONE-D-24-39301R1) submitted for publication in PLOS ONE.

We thank you for the thoughtful and positive feedback and constructive suggestions for improving the manuscript. We have revised the manuscript according to your recommendations. We have carefully considered and responded in detail to each of the points made by you and the reviewers, paying particular attention to clarity and detailing information on the relevant constructs in the introduction. As a result, we believe the paper is now stronger. In the following sections, we describe in detail how we have addressed each point. References not included in the manuscript, but cited in the reply are listed below the respective comment and its reply.

Reviewers' comments:

Reviewer #1:

This article addresses the role of positive and negative metacognitions in health anxiety and their relationship with pathological safety-seeking and avoidant behaviors. Extending previous findings from other anxiety-related disorders, such as generalized anxiety disorder (GAD), this systematic review seeks to bridge a critical gap in understanding metacognitive contributions to health-related anxiety. The meta-analysis of the results of 23 studies showed positive correlations between Negative (NMC) and Positive Metacognition (PMC) on one hand and health anxiety, as well as with safe seeking behaviors. No included study assessed avoidant behavior. While the methodology appears rigorous, with clear and well-detailed inclusion criteria for the selected studies, several parts of the paper lack clarity, particularly in the introduction. Additionally, inconsistencies in formatting and figures detract from the readability of the article.

Introduction

The introduction is thematically relevant but requires substantial clarification.

Comment 1. The primary research question is unclear, and some terms central to the study, such as "health anxiety", "safety-seeking behaviors", and "avoidant behaviors" would benefit from explicit definitions to ensure accessibility to a broader readership such as the one of PLOS ONE.

Reply: Thank you. Our primary research aim was to provide the first meta-analysis on the relation of metacognitions and health anxiety. We were interested in specific relations of positive and negative metacognitions with health anxiety, as well as the two classes of dysfunctional behavior typical for pathological health anxiety. Based on theoretical considerations, we assumed that positive metacognitions (e.g., “I can protect myself against getting a serious disease by thinking the worst about symptoms.”) might be more related with safety seeking behaviour and negative metacognitions (e.g., “I cannot control worrying on illnesses. It is overwhelming.”) might be more related with avoidance.

We detailed the information on the terms related to health anxiety (see also our answer to your Comment 3). In the updated manuscript, pathological health anxiety is defined as follows: “Pathological health anxiety is diagnosed when individuals are preoccupied with having or becoming a severe disease (15) and in its severe form a significant individual and societal burden in the long-term (16).” (lines 89-91)

Comment 2. Although the authors cite the metacognitive and cognitive-behavioral models, these frameworks are not adequately described, making it challenging for readers unfamiliar with these concepts to contextualize the study's objectives.

Reply: We detailed the information on the metacognitive framework: the cognitive-attentional syndrome is explicitly described, we more clearly distinguish between positive and negative metacognitions, clarify how dysfunctional coping strategies maintain worrying and negative metacognitions and provide a more detailed account of the empirical support for the model. The first part of the introduction is rephrased as follows (text parts with relevant changes are highlighted in yellow):

“Metacognitive beliefs refer to an individual's cognitive evaluations and beliefs about their own thoughts and thinking processes (e.g., 1). Maladaptive metacognitive beliefs play a significant role in the onset and exacerbation of various mental disorders (2,3). First being established for generalized anxiety disorder (GAD), Wells metacognitive model assumes that attentional biases, perseverative thinking (i.e., worrying and ruminating) and dysfunctional coping form a cognitive-attentional syndrome that manifests in mental disorders (4,5). In contrast to traditional cognitive models, thought process regulation by metacognitive beliefs lies at the heart of this framework, while thought content is considered as subordinate (i.e., the specific worry, for example ‘What if I lose my job?’). Contradictory positive metacognitions (PMC) and negative metacognitions (NMC) are held by people with pathological worries. PMC on the helpfulness of worrying initiate perseverative thinking. As an example, a person might think that worrying helps with problem solving, motivates, prepares for eventual catastrophes, or prevents harm. Whenever a specific worry is triggered, this person will continue to worry, because of those PMC. Engagement in perseverative thinking then activates NMC about the controllability and harmfulness of worries. When thoughts are experienced as particularly uncontrollable, people tend to engage in dysfunctional coping strategies, e.g., try to suppress their thoughts, or try to avoid triggers. As a result, they are more likely to remain stuck in perseverative thinking processes than people who attribute less importance to their thoughts. In other words, those strategies rather reinforce worrying and the perception of uncontrollability of worries (i.e., NMC). Various studies demonstrated that especially negative metacognitive beliefs relate to pathological worry in the context of GAD (5–8) and predict the duration, intensity, and burdensomeness of subsequent worry episodes (9,10), as well as generalized anxiety longitudinally (2).

In addition to their relevance for GAD, metacognitions are supposed to specifically impact mental disorders that are characterized by any kind of intrusive thoughts (11). In line with this, besides GAD, especially large effect sizes of NMC were found for obsessive compulsive disorder and major depression when contrasting different psychopathologies meta-analytically (2). Moreover, metacognitive therapy has been proven to be an effective transdiagnostic therapy, superior to standard cognitive behavioral therapy (for a meta-analysis see 12).” (lines 55-84)

Comment 3. The relationship between health anxiety, safety-seeking behaviors, and avoidance behaviors is not sufficiently detailed, leaving the reader without a clear understanding of how these constructs are conceptualized or interrelated.

Reply: Thank you for this important remark. We detailed the information on dysfunctional behaviors in illness anxiety as follows:

“Due to uncertainty concerning their health status, individuals may engage in excessive information sampling (e.g., checking the body, reassurance seeking, googling). Conversely, they might avoid any information that triggers their health anxiety (e.g., conversations about medical conditions, medical examinations, thinking about being ill). Safety seeking behaviors and avoidant behaviors are the two classes of dysfunctional behaviors typically maintaining pathological health anxiety. According to the Diagnostic and Statistical Manual of Mental Disorders (DSM 5), individuals can be categorized as either care-seeking or care-avoidant (15). However, most patients engage in both dysfunctional behaviors, and intra- and interindividual fluctuations remain poorly understood (17).” (lines 91-100)

Methods

The methodology is robust, and the explanation of study selection and bias assessment is clear. However, some areas might require additional detail:

Comment 4. How many studies reported regression coefficients and how were they transformed into correlation coefficients?

Reply: Only Bailey and Wells (2016) report regression coefficients. The three betas reported in this study were transformed into correlation coefficients using the esc_beta function (see uploaded Excel sheet and R script ‘2_RScript_MC-IA_datapreparation’). We have adjusted information in the manuscript as follows:

„Regression coefficients instead of correlation coefficients were reported in one study (33). The three extracted standardized regression coefficients were transformed into correlation coefficients with the ‘esc_beta’ function in R.“ (lines 203-205)

Comment 5. The section on data preparation mentions subgroups, what do they refer to?

Reply: Three of the studies (Barenbrügge et al., 2013; Bouman & Meijer, 1999; Fergus et al., 2022) report relevant correlation coefficients for different subgroups. Barenbrügge et al. (2013) distinguish between participants in ‘good health’ vs. in ‘poor health’ (whereby poor health was indicated by a self-reported doctor-diagnosed physical illness in the last three months, or one that persisted longer than the last three months). Bouman & Meijer (1999) categorize the sample into ‘hypochondriacal patients’, ‘normal controls’ and ‘psychology students’. And Fergus et al. (2022) differentiate between ‘non-latinx-black’, ‘latinx’ and ‘non-latinx-white’ subgroups. Information on the different subgroups as well as the corresponding correlation coefficients can be found in the uploaded Excel sheet. We provide one example for the subgroups in the section on data preparation and refer to the uploaded data and scripts as follows:

In the following citation, changed passages are highlighted in yellow: “For studies which used an instrument with more than one relevant subscale, or which reported the correlations separately for subgroups (k = 3, e.g., participants in good vs. poor health, 20), the corresponding correlation coefficients were averaged using Fisher’s z-transformation and weighted based on sample sizes (see uploaded data and scripts).“ (lines 205-209)

Comment 6. Figure 2 should include titles for panels A and B to improve interpretability.

Reply: In the title of Figure 2, we refer to both panels as follows: “Fig 2. Forest Plots for the relation of health anxiety and positive (A) as well as negative metacognitive beliefs (B).” We assume that the interpretability is clear once the manuscript has been set, the caption is placed directly above the Figure.

Comment 7. For analogue samples, it might be helpful to report the average levels and variability in health anxiety.

Reply: Thank you for this remark. We extracted reported levels of health anxiety for each study. We summarize the findings in the sample characteristics section:

“The average level of health anxiety varied between the studies ranging from low to severely health anxious on the respective self-report scales. Often, averaged health anxiety was mild. All mean values and standard deviations for the included studies can be found in the supplementary Table S5.” (lines 289-293)

and provide the detailed information in the supplementary Table S5.

S5 Table. Sample characteristics.

Average health anxiety scores of the included studies

Study N Question-naire Possible range of scores in the respective question-naires Average Health Anxiety Score M (SD) Interpretation of Averaged Health Anxiety Score Sample

Melli et al. 2018 (1)

458 HAQ 21 to 84 64.69 (10.78) - clinical

Melli et al. 2016 (2)

342 HAQ 21 to 84 32.47 (8.70) - analogue

Airoldi et al. 2022 (3)

125 SHAI-14 0 to 42 25.69 (6.08) above cut-off, mild analogue

Nadeem et al. 2022 (4)

500 SHAI-14 0 to 42 18.72 (8.77) above cut-off, mild analogue

Akbari et al. 2022 (5)

541 SHAI 0 to 54 29.74 (6.13) above cut-off analogue

Dai et al. 2018 (6)

1191 SHAI 0 to 54 11.73 (5.32) below cut-off analogue

Penney et al. 2020 (7)

565 SHAI 0 to 54 16.47 (8.18) above cut-off analogue

Rachor & Penney 2020 (8)

179 SHAI 0 to 54 20.27 (7.46) above cut-off analogue

Bailey & Wells 2016 (9)

105 WI (Likert) 14 to 70 26.36 (9.36) below cut-off analogue

Bailey & Wells 2013 (10)

351 WI (Likert) 14 to 70 26.37 (10.14) below cut-off analogue

Bailey & Wells 2015 (11)

235 WI (Likert) 14 to 70 not reported analogue

Barenbrügge et al. 2013 (12)

1264 WI (dichotome) 0 to 14 3.39 (2.87) below cut-off analogue

Bouman & Meijer 1999 (13)

normal controls 25 WI (dichotome) 0 to 14 2.0 (2.0) below cut-off analogue

psychology students 122 WI (dichotome) 0 to 14 2.7 (2.2) below cut-off analogue

hypochondrial patients 14 WI (dichotome) 0 to 14 10.9 (2.2) above cut-off clinical

Kaur et al. 2011 (14)

158 WI (Likert) 14 to 70 13.94 (9.00) below cut-off analogue

Solem et al. 2015 (15)

382 WI (Likert) 14 to 70 23.3 (7.70) below cut-off analogue

Fergus & Bardeen 2019 (16)

785 WI-6 6 to 30 not reported - analogue

Fergus & Spada 2017 (17)

337 WI-6 6 to 30 13.14 (7.68) - analogue

Fergus & Spada 2018 (18)

Study 1 330 WI-6 6 to 30 13.66 (5.20) - analogue

Study 2 331 WI-6 6 to 30 13.7 (5.34) - analogue

Fergus et al. 2022 (19)

non-latinx black 123 WI-6 6 to 30 14.53 (5.95) - analogue

latinx 104 WI-6 6 to 30 14.53 (5.46) - analogue

non-latinx white 80 WI-6 6 to 30 15.71 (6.72) - analogue

Zheng et al. 2021 (20)

426 WI-6 6 to 30 not reported - analogue

Cartwright-Hatton & Wells 1997 (21)

47 AnTI 6 to 24 9.5 (3.8) - analogue

Wells & Papageorgiou 1998 (22)

105 AnTI 6 to 24 9.4 (not reported) - analogue

Abbreviations: HAQ = Health Anxiety Questionnaire, note response format of the Italian version used differs from the English version (shifting of the scale values), determined by personal correspondence with Gabriele Melli, Melli et al. (29) report average values of the Italian HAQ version of: M = 31.6, SD = 7.9 for men and M = 36, SD = 10.6 for women, HAI-14 = Short Health Anxiety Inventory-14 (14 Items), note a cut-off of 18 was suggested for analogue samples (‘healthy context’) and values are interpreted as follows: 0-27: no to mild health anxiety, 28-32: moderate health anxiety, 33-42: substantial health anxiety (23), SHAI = Short Healthy Anxiety Inventory (18 Items), note a cut-off of > 14 is interpreted as noticeable health anxious (24), WI = Whiteley Index (14 Items), note the cut-off depends on the response format (dichotome vs. 5-point Likert-scale), for WI (dichotome) a cut-off of 5 was used to discriminate persons with severe health anxiety from those without severe health anxiety (25) and for WI (Likert) a cut off score of 40 or above indicates the presence of hypochondriasis (26), WI-6 = Whiteley Index (6 Items), note Fergus et al. (27) report M = 21.28, SD = 5.54 for severe health anxiety and M = 13.49, SD = 5.60 for no severe health anxiety, there is no cut-off criterion, AnTI = Anxious Thought Inventory: the subscale health worry was used to determine the level of health anxiety, note Vázque Morejón et al. (28) report M = 12.66, SD = 4.84 in a clinical sample

References

1. Melli G, Bailey R, Carraresi C, Poli A. Metacognitive beliefs as a predictor of health anxiety in a self‐reporting Italian clinical sample. Clin Psychol Psychother. 2018 Mar;25(2):263–71.

2. Melli G, Carraresi C, Poli A, Bailey R. The role of metacognitive beliefs in health anxiety. Personal Individ Differ. 2016 Jan 1;89:80–5.

3. Airoldi S, Kolubinski DC, Nikčević AV, Spada MM. The relative contribution of health cognitions and metacognitions about health anxiety to cyberchondria: A prospective study. J Clin Psychol. 2022 May;78(5):809–20.

4. Nadeem F, Malik N, Atta M, Ullah I, Martinotti G, Pettorruso M, et al. Relationship between Health-Anxiety and Cyberchondria: Role of Metacognitive Beliefs. J Clin Med. 2022 May 5;11(9):2590.

5. Akbari M, Spada MM, Nikčević AV, Zamani E. The relationship between fear of COVID‐19 and health anxiety among families with COVID‐19 infected: The mediating role of metacognitions, intolerance of uncertainty and emotion regulation. Clin Psychol Psychother. 2021 Nov;28(6):1354–66.

6. Dai L, Bailey R, Deng Y. The reliability and validity

---

## [Decision Letter · Decision Letter 1]

Do metacognitions contribute to pathological health anxiety? A systematic review and meta-analysis.

PONE-D-24-39301R1

Dear Dr. Pohl,

We’re pleased to inform you that your manuscript has been judged scientifically suitable for publication and will be formally accepted for publication once it meets all outstanding technical requirements.

Kind regards,

Hans-Peter Kubis, PD. Dr. rer. nat.

Academic Editor

PLOS ONE

Additional Editor Comments (optional):

Reviewers' comments:

Reviewer's Responses to Questions

**Comments to the Author**

Reviewer #2: All comments have been addressed

2. Is the manuscript technically sound, and do the data support the conclusions?

Reviewer #2: Yes

3. Has the statistical analysis been performed appropriately and rigorously?

Reviewer #2: Yes

4. Have the authors made all data underlying the findings in their manuscript fully available?

Reviewer #2: Yes

5. Is the manuscript presented in an intelligible fashion and written in standard English?

Reviewer #2: Yes

Reviewer #2: (No Response)

**Do you want your identity to be public for this peer review?** For information about this choice, including consent withdrawal, please see our Privacy Policy

Reviewer #2: No

---

## [Editor Report · Acceptance letter]

PONE-D-24-39301R1

PLOS ONE

Dear Dr. Pohl,

I'm pleased to inform you that your manuscript has been deemed suitable for publication in PLOS ONE. Congratulations! Your manuscript is now being handed over to our production team.

Kind regards,

on behalf of

Dr. Hans-Peter Kubis

Academic Editor

PLOS ONE